# A Needs Assessment Approach for Adolescent and Young Adult Sexual and Gender Diverse Cancer Survivors

**DOI:** 10.3390/ijerph21040424

**Published:** 2024-03-30

**Authors:** Lauren V. Ghazal, Hailey Johnston, Elisabeth Dodd, Yasmine Ramachandra, Nicholas Giallourakis, Kayla Fulginiti, Charles Kamen

**Affiliations:** 1School of Nursing, University of Rochester, Rochester, NY 14642, USA; 2Wilmot Cancer Institute, Rochester, NY 14642, USA; charles_kamen@urmc.rochester.edu; 3Escape, Lansing, MI 48915, USA; hnjohnst@umich.edu (H.J.); elisabethcdodd@gmail.com (E.D.); yasmine.ramachandra@gmail.com (Y.R.); 4School of Public Health, University of Michigan, Ann Arbor, MI 48109, USA; 5Elephants and Tea, Avon Lake, OH 44012, USA; nick@elephantsandtea.com (N.G.); kayla@elephantsandtea.com (K.F.); 6Department of Surgery, Cancer Control, University of Rochester Medical Center, Rochester, NY 14642, USA

**Keywords:** sexual and gender diverse, sexual and gender minority, LGBTQ+, support, mental health, community organization, community engagement, needs assessment

## Abstract

Sexual and gender diverse (SGD) adolescent and young adult (AYA) cancer survivors are an increasing and vulnerable group with unique needs that often remain unmet in the healthcare system. This paper describes the conceptualization and development of a community-based organization dedicated to serving SGD AYAs, in addition to reporting on the results of a community-led needs assessment. A total of 56 SGD AYA community members completed the online survey. Most participants were between the ages of 26 to 33, identified as white, cisgender, bisexual women, and had hematologic malignancies. Identified unmet needs of SGD AYAs included the following: sexual health and family planning; gender affirmation; financial stability; and emotional support. Areas within the community organization were identified as gaps, areas of expansion, and assets. Results highlight the role of community and academic partnerships in improving cancer care delivery for SGD AYA cancer survivors.

## 1. Introduction

The intersections of cancer survivorship (i.e., from time of diagnosis to end of life) and sexual and gender diverse (SGD) status for adolescents and young adults (AYAs) pose significant challenges that result in cancer-related health and healthcare disparities [1,2]. Sexual and gender diverse (also described as “sexual and gender minority”) status is used in this paper as the umbrella term for those who identify as lesbian, gay, bisexual, asexual, transgender, Two-Spirit, queer, and/or intersex (LGBTQI2S+) [3]. While SGD populations overall are historically understudied, the increasing population of SGD AYA cancer survivors warrants specific attention to this distinct age group to address unmet cancer-related needs.

In 2023, it is estimated that nearly 86,000 AYAs (15–39 years old) were diagnosed with cancer in the United States (US) [4]. From 2010 to 2019, cancer incidence increased by an average of 0.3% each year, with, most recently, a marked rise in adult-onset cancers in the AYA population [5,6]. At the same time, the general population of AYAs represents the largest growing population of SGD individuals [7]. However, current epidemiologic data lack the ability to sufficiently capture the prevalence of SGD AYA cancer survivors in the US [8]. One reason is that sexual orientation and gender identity (SOGI) data collection is still not standardized across not only cancer care settings but healthcare delivery settings in general [9]. In 2023, polling and focus groups conducted by the Public Religion Research Institute (PRRI) estimated that 28% of Generation Z (born 1997–2004) and 16% of Millennials (born 1981–1996) identify as SGD [10].

SGD AYAs experience disparities in cancer care delivery that are driven by unmet cancer-related needs [11,12]. A 2023 scoping review of disparities among SGD AYAs identified numerous gaps in cancer care and outcomes research [1]. Prior studies have identified the multiple healthcare and psychosocial challenges that SGD cancer patients face, compared with their non-SGD counterparts. These challenges include higher levels of psychological distress, greater financial burden, and additional barriers to receiving cancer care [13,14,15,16]. Data from the National Health Interview Survey, for example, found that lesbian, gay, and bisexual (LGB) AYAs were at increased risk of having chronic health conditions compared with both LGB individuals without a history of cancer and heterosexual AYAs [17].

SGD AYAs possess unique intersectional identities of SGD status and cancer as an AYA [1,18,19]. These experiences alone can lead to instances of extreme vulnerability and isolation. Compiled, these experiences can present challenges across one’s cancer survivorship and process of “coming out”. That is, throughout this period of emerging adulthood and young adulthood, AYA cancer survivors could have “come out” prior to their diagnosis, they could be in the process of coming out amid their diagnosis, or they could “come out” following their diagnosis and treatment. SGD communities have historically sought refuge and safety in chosen families or communities, especially in the context of health and illness and receipt of healthcare [20]. Yet, there remain questions about SGD AYAs in the context of cancer and what specific organizational support is available in community settings. The purpose of this manuscript is to describe the findings of the community-led needs assessment, identify the organizational support response to report findings, and report on the conceptualization and development of a community-based organization specifically serving SGD AYA cancer survivors.

## 2. Materials and Methods

### 2.1. Study Design

In partnership with community and academic organizations and AYA patients, we conducted an exploratory investigation into the experiences of SGD AYAs with cancer, with the goal of informing the organizations’ work and how to best serve the SGD AYA community. Our work was guided by the community-based participatory research (CBPR) framework [21], which highlights the interplay among community members and the research process. Together, we engaged in co-learning and bidirectional transferring of expertise and knowledge, recognizing that all members of our team were research partners [22,23,24], and many were embodied researchers as patient scientists and/or SGD individuals [1,25,26]. Institutional Review Board oversight was not required for this study. The data were not linked to identifiers and are not considered human subjects research.

### 2.2. Setting

Two community-based organizations collaborated on this needs assessment. Escape (escapeayac.org; accessed on 10 January 2024) led the assessment, and Elephants and Tea (elephantsandtea.org; accessed on 10 January 2024) supported the effort in authentic allyship and collaboration. The collaboration was founded by the organizations’ shared values of creating space in cancer care that is safe for self-expression and building community through centering the patient’s voice.

Escape and Elephants and Tea: Escape was founded in 2018 by SGD AYA cancer survivors and was created to highlight the voices of SGD AYA cancer survivors [27]. The naming of the organization was very intentional and accounted for the safety of its members by incorporating SGD-specific language in the title. As an empowerment-based organization working to provide a literal “escape” from all that comes with going through cancer as an SGD person, Escape fosters a goal of developing a haven for SGD AYAs where they can be in community with peers who share their experience on a deeper level.

Founded in 2018, Elephants and Tea is a non-profit media brand of the Steven G. Cancer Foundation [28]. Their mission is to help AYA patients, survivors, and caregivers know they are not alone in their experience with cancer. They are the only magazine written for and by the AYA cancer community committed to telling the stories of this unique and diverse group. Their name signifies that cancer is the “elephant in the room” and “tea” is the relief that conversation and storytelling provide.

### 2.3. Data Collection

From December 2021 to June 2022, we distributed an online survey through the community organizations’ social media accounts and community listservs. The survey was housed on SurveyMonkey (surveymonkey.com; accessed on 3 June 2023). The 22-question survey comprised multiple-choice and open-ended questions that were investigator-designed and co-developed with community input. These investigators had expertise in SGD health, AYA cancer survivorship, and trauma-informed cancer care. Questions captured demographic information, such as age, sexuality, gender, race, relationship to cancer, and diagnosis. Additionally, open-ended questions were asked to gather qualitative data on experiences accessing healthcare as LGBTQI2S+ individuals. The methodology allowed for the collection of data from a diverse range of participants and gathered both quantitative and qualitative data to provide a comprehensive understanding of the experiences of SGD AYA cancer patients.

Included questions related to community support and self-care as an SGD community member. For example, questions included “What does self-care look like for you?” and “What are some pieces of feedback that would improve your experience with healthcare providers? And what does community look like for you? How specifically can Escape foster that community?” Eligible participants were >18 years old at the time of study completion and an SGD AYA cancer community member. An SGD AYA cancer community member was defined by the community organizations as at least one of the following: (1) personally identifying as SGD and an AYA cancer survivor; (2) providing care to an SGD cancer survivor (e.g., caregiver); and/or (3) being a healthcare provider in the space. Ineligible participants included individuals under the age of 18, individuals diagnosed over the age of 39, and individuals who had experienced a cancer diagnosis but did not personally identify as an SGD AYA.

### 2.4. Data Analysis

Survey responses were de-identified by one community organization and shared with the larger investigative team on a password-protected document. Quality data assurance was performed by two members of the study team, and we calculated descriptive statistics on demographics and multiple-choice questions.

We then performed a directed content analysis on the short and long responses to generate categories of unmet needs [29,30]. This process involved frequent meetings both via Zoom and in person when able to discuss any disagreements and assess if the goals for moving the research along with this needs assessment were still in alignment with the goals of the organizations. Two of the authors (HJ and ED) initially coded the data and secondary coding was performed by another author (LG). Qualitative data saturation was defined as met when no new codes emerged from the data. Following initial coding, general categories of codes were discussed and combined into themes.

Following a review of the data with community members, the investigative team reviewed the mission and vision of the leading organization. In this feedback loop, Escape applied Simon Sinek’s Golden Circle of Theory Value Proposition to assist in reframing their organizational vision and resources provided [31,32]. The Golden Circle of Theory Value Proposition highlights how great leaders and organizations should plan their organizational mission from the inside out of this circle—first starting with their organization’s “why”, and then moving outward of the circle to “how” they will get there and “what” it is they will do.

## 3. Results

A total of 56 participants completed the survey (Table 1). Most participants were cis-gender women (58.9%). Non-binary participants comprised nearly 20% of the sample, and participants who were either transgender men or women represented 7.2% of the sample. Sixty-one percent were SGD AYA cancer survivors themselves, 12.5% were caregivers, and nearly 18% were healthcare or non-profit professionals. Thirty-two percent were aged 26–33 years, 20% were 18–25 years, and 20% were between the ages of 34 and 39. As a result of looking across the SGD AYA community of caregivers and providers, some participants ranged in ages of >40 years. Nearly 57% of participants were white, and 20% were Hispanic or of Latin or Spanish origin. Nearly 9% of participants were multiracial or biracial. Nearly 9% of participants were Asian, Pacific Islander, or Middle Eastern.

Regarding SGD status, most respondents were cisgender women (59%). Roughly 20% identified as non-binary, 2% as transgender women, and 5% as transgender men. Roughly 23% identified as bisexual, 20% as gay, 18% as lesbian, and 18% as queer, with 5% of participants identified as straight (Table 1).

Participants provided an open-ended response detailing their cancer diagnosis(es), which led to a wide range of information. Some responses were detailed and included specific information such as stage and subtype, while others were more general and only indicated the broad category of the cancer type. The majority were hematologic malignancies (leukemia, lymphoma) (40%), breast (13.8%), and gynecologic cancers (13.8%). Some participants noted multiple diagnoses; however, their diagnoses all fell under the same cancer type category.

### 3.1. Unmet Needs

Participants described four major unmet needs related to SGD AYA cancer care: (1) sexual health and family planning; (2) gender affirmation; (3) financial stability; and (4) emotional support (Table 2). Participants endorsed significant gaps in provider education related to sexual health and family planning, which included the perceived comfort or discomfort level of the provider: “*I had such a great team of providers but sexual health is not an easy topic to address with your doctors especially if that is not their specialty*”. Participants also noted needs around gender affirmation, where one participant described how treatment-related effects coincided with provider conversations around sexual orientation and gender identity: “*I was actually excited to lose my hair and potentially my fertility! It would mean less pressure for me to perform femininity*”.

Lastly, financial needs were intricately linked to mental health needs and emotional support, where participants described the disturbance of having to ask for financial support, or the sensitivity of financial issues that presented with mental health therapy.

### 3.2. Conceptualizations of Community, Support, and Self-Care

The needs assessment also illuminated participant’s conceptualizations of community, support, and self-care. Overall, participants described “community” as family/friends/chosen family being supportive, a safe space free of judgment, a feeling of belonging, and the sharing of resources and mutual aid. Many identified family and friends as a support system, and respondents included biological and “chosen family”, which they defined as SGD peers, friends, colleagues, and members of their medical team. Many shared that their support system was supportive, but that it was hard because they did not share the same intersectional identities of cancer and SGD identity and could not relate and understand their distinct experiences. Others reported receiving support from family that did not fully accept their queer identity. Some respondents also described having no support from family and having to find and lean on their “chosen family”. Peer connection and creating a safe space also highlighted the importance of storytelling.

Participants described ways they had experienced community support throughout their diagnosis and treatment, which resulted in participants feeling not alone because they received check-ins through messages, visits, emails, and cards. Additional examples included the sharing of memes, having Zoom meetings to check in, and hosting virtual and in-person game nights. Others included emotional support listening, practicing empathy, and listening to understand and not just giving their advice. Participants also endorsed support while processing their SGD identity and shared that affirming their authentic selves was an opportunity for community support. Several participants shared that they do not have community support, and offered practical ways for their community to support them best. This included shared ideas for social gatherings, such as local events and groups like a sports league, and events through local centers. Participants also acknowledged that financial support for mental health services could also be beneficial to support them.

When conceptualizing self-care as SGD AYA cancer survivors, participants also described avoidant coping and distraction. Perspectives of self-care presented differently when compared to perspectives of escapism. Participants’ responses to self-care examples included: helping others/getting involved, dressing in gender-affirming clothes, self-compassion, living authentically, and setting boundaries. Some participants also reported not exactly “know[ing] how to take care of [themselves]”. However, participants were able to describe how they were able to distract themselves (escapism) and included things like social media. That is, self-care was more often described as activities in the community with energy, while escapism presented individually, and when they did not have the energy.

### 3.3. The Role of Escape: Gaps, Areas of Expansion, and Assets

Gaps, areas of expansion, and assets in relation to quality of care were found in the services that Escape provides and in addressing the needs of SGD AYA cancer survivors (Table 3 and Figure 1). Within these three categories (gaps, areas of expansion, and assets), twelve areas were identified. Participants predominantly expressed gaps in provider communication, emotional support, mental health care access, and LGBTQI2S+ affirmation. Participants also reported gaps in SGD and BIPOC representation in cancer-related media, relevant resources, and SGD sexual health as it relates to cancer. Many identified a lack of sustaining SGD cancer support groups and the need for peer connection both locally and virtually.

The role of community-based organizations, such as Escape, was highlighted in the actions they can take with, and *for*, the SGD AYA community. Several respondents who had interacted with Escape before shared that the monthly meet-ups and the connections that they had made were very supportive. Participants who initially discovered Escape through the needs assessment survey shared that they were excited to have found this organization because there are so few SGD AYA-specific organizations. Several participants stated that they would be keeping an eye out for Escape events in the future.

The question “*What would you like to know more about as someone in the LGBTQI2S+ community?*” provided more than twenty topic areas that Escape could support through educational programming. Participants could select all they felt interested in and there was also an option to write in other topic areas that may have not been listed. Participants predominantly reported wanting to know more about systems navigation as an AYA, self-advocacy, communication with healthcare providers, financial planning, gender-affirming oncology care, sex and intimacy, and legacy planning as transgender and gender-diverse people.

Across all response data, provider communication was the dominant concern raised by participants (Figure 1). This included a provider’s ability to actively listen, trust patient experiences, and clearly share expectations as facets of communication. Participants reported the need for their providers to be more mindful of the social and mental impacts of cancer in their approach to treating the diagnosis. Participants also expressed the need for cultural humility in how they are provided care. Examples included asking for and using the personal pronouns of patients, leading with curiosity instead of assumptions, having a baseline knowledge of sexual health and gender-affirming care as it relates to SGD people, and safety considerations such as hospital room assignments with individuals who do not hold prejudice toward SGD people.

## 4. Discussion

This needs assessment reports unmet needs related to SGD AYA cancer care by asking SGD AYA community members directly. The survey disseminated through community organizations provides valuable insights into the experiences and needs of SGD AYAs with cancer and highlights the importance of inclusive and tailored support for this population. This needs assessment is a collective effort, guided by underpinnings in community-based participatory research, to learn how best to serve SGD AYAs through community-based organizations and advocacy efforts.

Little literature currently exists from community organizations conducting assessments on the experience of SGD AYAs in cancer care. However, the findings of this needs assessment are consistent with the existing research surrounding SGD experience in healthcare and oncology. The lack of an SGD focus being adopted in service provider training is illustrated in the participant response data on gaps in provider communication and complements research that has been conducted on the representation of SGD people in medical assessment and teaching scenarios [33]. The needs assessment findings also build upon research on SGD experience in oncology conducted by the National LGBT Cancer Network through their OUT survey in 2020. Findings expand on themes related to adolescents and young adults, community and familial support, access to care, the transgender experience in oncology, and mental health [34].

### 4.1. Post-Needs Assessment Evaluation

#### 4.1.1. The Why

In determining where additional support is needed among SGD AYAs, it was important for Escape to apply a critical lens in reassessing their mission identifying the starting question of why it is they exist (Figure 2). In doing this, they determined that the purpose of the organization is to fill the gaps in supportive resources for SGD AYAs and deliver these resources through an intersectional, person-centered, and trauma-informed approach [35]. Escape also provides education to institutions and community organizations working to better serve SGD AYAs and acts as a partner in addressing the unique needs of this population in oncology.

#### 4.1.2. The How

As an organization entirely led by individuals with lived experience being SGD and personally impacted by cancer at an AYA, a key priority of Escape is to decrease isolation among SGD AYAs and improve health outcomes among those diagnosed with cancer through health-promoting support and interventions. Escape executes this work in a unique way compared to any other organization that serves the AYA population. First, Escape has intentionally excluded SGD-related language in the organization title and branding to create a layer of safety for those who may not be “out” to their caregivers. Since Escape exclusively serves the SGD population, this allows them to create and promote events and communicate with participants without explicitly using SGD language. Through this intentionality of the organization structure, Escape has become a haven for SGM AYAs who may be in unsafe situations with caregivers who do not affirm their identity.

Second, Escape is the only AYA cancer support organization that exclusively serves SGD AYAs internationally. There are some AYA organizations that offer support groups for this population and SGD cancer organizations that serve all ages, but there is no other resource available that can prioritize anonymity in the same way as Escape, thus making Escape not only unique in the non-profit space, but a vital contributor to this area of supportive care in AYA oncology.

The goal of Escape is to cultivate a space where SGD AYAs feel safe, represented, heard, empowered, and amplified through meaningful community collaboration. Elephants and Tea has been a close partner with Escape in furthering their mission goals and visibility as an organization by promoting the organization in their quarterly magazine and partner resource lists, promoting all Escape events on their programs and events webpage, and as a participant in Escape’s annual LGBTQIA+ Cancer Awareness Week campaign during June. Financial support from Elephants and Tea was also provided in the research infrastructure needed to disseminate the needs assessment survey. Through this partnership, Escape has been able to integrate quickly into the AYA oncology support space and build capacity as an organization in a way that would not have been possible without this partnership.

#### 4.1.3. The What

When first forming this partnership with Elephants and Tea, all Escape had was a website that allowed people to find the organization’s Facebook support group and other social media accounts, and they would occasionally host peer meetups over Zoom. Now, Escape has become the leader in serving the AYA SGD population through regular virtual programming for psychosocial support, an annual campaign during the month of June, opportunities for mutual aid, equitable and affirming resource guides, and educational resources for the community. Much of this growth since 2020 has been made possible through Elephants and Tea amplifying Escape’s work and connecting its board of directors with other partners to help build capacity and further Escape’s mission. Active aspiring allyship modeled by Elephants and Tea has been crucial in Escape’s success as an organization. Elephants and Tea are leaders in AYA narrative storytelling and through their opportunities for Escape to share their story and mission, they have given voice to a population who has been entirely excluded from the conversation in AYA oncology. Furthermore, as an organization that is seen as a pillar of support in the AYA oncology space, their solidarity with the Escape community has created a layer of safety for the work Escape does in supporting SGD AYAs.

### 4.2. Future Goals and Implications

There are actions that Escape can take with not only AYA cancer survivors, but also oncology professionals and other community organizations. Board members wanted to ensure that Escape was growing intentionally in a direction that truly serves the varied needs of the community. To do this, it was necessary to establish what community means to the people Escape is trying to serve. The findings from this study will create a foundation for the services Escape provides and a blueprint for how to close gaps in care for the SGD AYA population. Escape plans to do this on an individual, community, and systemic level through patient and caregiver advocacy, events, and provider education.

Most recently, Escape received grant funding from the Trans Justice Funding Project (https://www.transjusticefundingproject.org/; accessed on 3 February 2024) to create a Trans AYA Toolkit and webinar series. Many survey respondents emphasized the importance of self-advocacy and acknowledged difficulty doing this in a medical setting. There are many barriers to accessing services as an SGD AYA. Escape hopes that this toolkit will equip patients and caregivers with the knowledge they need to anticipate roadblocks so that people can effectively navigate issues as they arise. This toolkit will include the concerns brought up by respondents including information about gender-affirming care and cancer treatment, an SGD resource directory, self-care coping strategies, and practical guides for discussing options and concerns with medical care providers. With this information, folks can feel empowered to ask for what they need.

Currently, Escape facilitates two monthly peer support spaces over Zoom. People who have attended shared that this virtual community space has been invaluable while also expressing a desire for more structured events. With this feedback, Escape plans to incorporate more group activities into its monthly programming, with clear activities and agendas.

In our study, we identified two prominent issues that participants reported: poor communication and lack of patient-centered care among SGD AYAs. It has always been the goal for Escape to train medical providers, researchers, and cancer advocates on SGD service provision. Analysis of survey responses was helpful in identifying specific skills and education, and Escape plans to create a curriculum that will help medical providers ground their communications in cultural humility and trauma-informed care, educate them on the unique support needs of SGD AYAs, and emphasize the importance of trusting a patient’s experience while letting them lead [35].

Overall, this study has important implications for research, not only for SGD AYA populations, but for larger community-based participatory research. Results from this needs assessment and reflective contributions of a community-based organization can help to foster research partnerships integral for funding opportunities, such as the National Cancer Institute’s funding opportunity, Improving Care and Outcomes for Cancer Survivors from Sexual and Gender Minority (SGM) Populations [36]. By disseminating our findings, we hope to facilitate a broader dialogue and encourage the development of more inclusive and accessible resources for the SGD AYA cancer community.

### 4.3. Limitations

While the results of our community survey of SGD cancer patients provide valuable insights into the experiences and needs of this population, it is important to acknowledge the limitations of our research. One major limitation is the small sample size of only 56 participants, which may not be representative of the larger SGD AYA population. However, the sample represented in this study does capture the prevalence of commonly occurring diagnoses in this age range, including hematologic malignancies. Additionally, the outreach for the survey was conducted through cancer non-profits, which may have biased the sample towards individuals who already had access to some form of psychosocial support. Therefore, caution should be taken when generalizing the findings to the larger SGD cancer patient population, and future research should aim to recruit a more diverse and representative sample. Lastly, the study survey was shared in 2021 when there were protocols in place to reduce transmission of COVID-19. Many of these precautions prevented larger in-person gatherings. This could contribute to the isolation that people were feeling, and the desire to connect with others.

## 5. Conclusions

One size does not fit all in cancer survivorship research. There is a critical need to amplify the experiences of diverse AYA cancer survivors, and, importantly, SGD AYAs. Results from the needs assessment highlight the gaps (unmet needs) in care for the SGD AYA cancer population and show how community organizations can begin to address specific efforts needed to change the landscape of cancer care for the SGD AYA cancer community. Both organizations (Escape and Elephants and Tea) are committed to efforts to promote health equity in AYA oncology by amplifying the voices of those underrepresented in the AYA cancer community through peer support and storytelling. Collaborations are critical to inclusive healthcare that is safe for self-expression.

## Figures and Tables

**Figure 1 ijerph-21-00424-f001:**
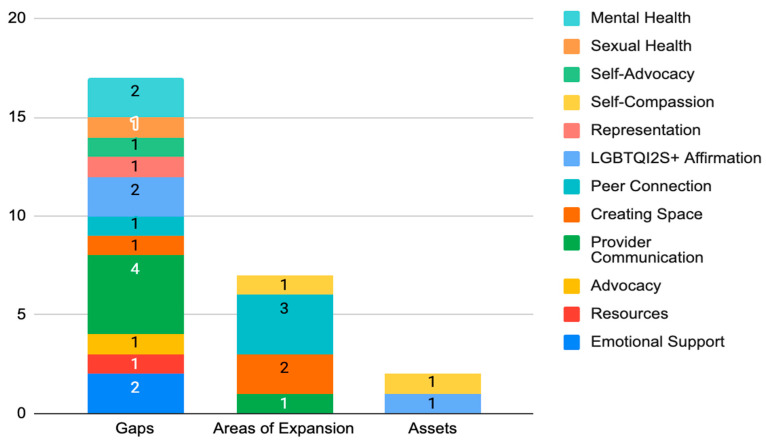
Gaps, expansion areas, and assets of Escape. This stacked bar chart depicts the targeted areas that were identified as gaps, areas of expansion, and assets based on participant response data. This chart highlights the presence of significantly more gaps communicated than both areas of expansion and assets. The numbers depicted in the stacked bars represent the frequency of specific gaps, areas of expansion, and assets.

**Figure 2 ijerph-21-00424-f002:**
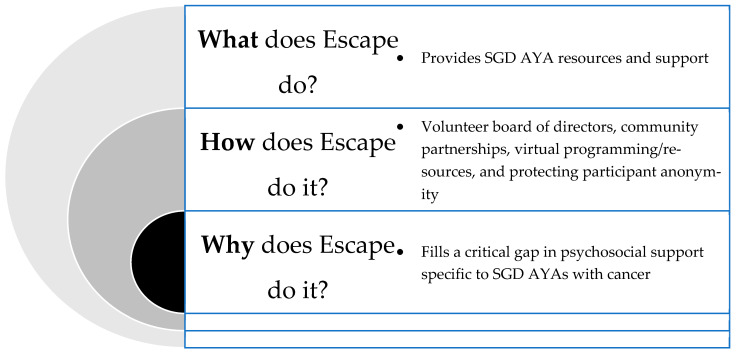
Using Sinek’s Golden Circle of Theory Value Proposition, we frame Escape’s response to the needs assessment findings in the context of the why, how, and what [31,32].

**Table 1 ijerph-21-00424-t001:** Participant sociodemographics (*N* = 56).

Characteristic	*n* (%)
**Role ^a^**	
Patient	31 (55.4)
Survivor	36 (64.3)
Healthcare professional	8 (14.3)
Caregiver/supportive loved one	7 (12.4)
Non-profit professional	2 (3.6)
Prefer not to disclose	1 (1.8)
**Age (years)**	
18–25	11 (19.6)
26–33	18 (32.1)
34–39	11 (19.6)
40–45	6 (10.7)
46–49	2 (3.6)
50–60	6 (10.7)
61+	2 (3.6)
**Race/Ethnicity**	
White or Caucasian	32 (57.1)
Hispanic or Latino or Spanish origin of any race	11 (19.6)
Multiracial or Biracial	5 (9.0)
Middle Eastern	3 (5.4)
Black or African American	2 (3.6)
White or Caucasian: Ashkenazi Jewish	1 (1.8)
Asian or Pacific Islander	1 (1.8)
Prefer not to disclose	2 (3.6)
**Gender Identity**	
Cisgender woman	33 (58.9)
Cisgender man	5 (8.9)
Non-binary	11 (19.6)
Transgender man	3 (5.4)
Transgender woman	1 (1.8)
Prefer not to disclose	3 (5.4)
**Sexual Orientation ^a^**	
Asexual	3 (5.4)
Bisexual	13 (23.2)
Lesbian	10 (17.9)
Gay	11 (19.6)
Graysexual	1 (1.8)
Pansexual	5 (8.9)
Queer	10 (17.9)
Questioning	3 (5.4)
Sapiosexual	1 (1.8)
Straight	3 (5.4)
**Cancer Diagnosis**	
Blood	24 (41.4)
Brain	4 (6.9)
Breast	8 (13.8)
Gynecologic	8 (13.8)
Melanoma	3 (5.2)
Sarcoma	2 (3.4)
Other ^b^	10 (17.1)

^a^ May total more than 100%, as participants could select more than one option. ^b^ Includes Adrenal, Bladder, Kidney, Myeloma, Myoepithelial Carcinoma, Neuroendocrine, and Testicular.

**Table 2 ijerph-21-00424-t002:** Unmet needs among SGD AYA cancer survivors.

Need	Exemplary Quotes
Sexual Health and Family Planning	“Need more sexual health resources… like how to have safe sex as a [queer and transgender] immunocompromised person”. “I wish there could have been an LGBTQ+ physician or care provider that I could’ve had as a liaison to direct any questions about my mental or sexual health to. I had such a great team of providers but sexual health is not an easy topic to address with your doctors especially if that is not their specialty. I feel like health systems don’t educate patients of the LGBTQ+ community with relevant sexual health information”. “I wish they spoke to all gyn cancer patients about sex, sexuality, gender, and pelvic floor physical therapy, surgical menopause, and what to expect (i.e., what typical recovery might be like versus what the ‘new normal’ might be like). They spoke to me about fertility preservation (and then that specialist determined I was ‘too high risk’, which made me feel worse). Asking these questions felt impossibly hard, and when I tried, the replies I received were as though I wasn’t being grateful for living, or as though my concerns were frivolous or irrelevant since I’m not married [to a man]”.
Gender Affirmation	“I wish someone would have talked to me about the difficulties I would face, having a “female” cancer but a gender-neutral to masculine presentation, and how to speak up for myself and also care for my own emotions around my identity”. “Trans people have a harder time being affirmed by their oncologists and our struggles are not the same as cis people”. “I was actually excited to lose my hair and potentially my fertility! It would mean less pressure for me to perform femininity”.
Financial Stability	“Our experience is unique—be sensitive to the fact we deal with trauma before many other groups in the community. We are more likely to have financial trouble—be sensitive to helping.” “Financial issues are real. Wish therapy was cheaper”. “Honestly, financial support is really important. Hate to ask for it”.
Emotional Support	“After coming out as Pan and transitioning to palliative, care-I have lost much support. Community support looks like a place I can go and be safe, share my feelings, and not be judged”. “I am very lucky with how supportive my family (and some friends) has been with my cancer. They are not fully on board with my sexuality but they are not fully against it either”.

**Table 3 ijerph-21-00424-t003:** Assessment of Escape’s gaps, areas of expansion, and assets in relation to SGD AYAs.

Guiding Question(s)	Area	Gap, Area of Expansion, or Asset
How can Escape foster community?	Peer Connection, Creating Space	Expansion
If you have community support, how can they show up for you?	Emotional Support	Gap
How can your community support you best?	Creating Space, Self-advocacy	Gap
What does self-care look like for you as an LGBTQI2S+ identifying individual?	Self-compassion, LGBTQI2S+ Affirmation	Asset
What has been your escape during your cancer experience?	Self-compassion	Expansion
What would you like other LGBTQI2S+ community members to know beforehand when seeking cancer care? What do you wish was shared with you?	Self-advocacy, Resources	Gap
Is there anything you wish your medical team had gone over with you which they didn’t?	Sexual Health, Mental Health	Gap
How is the cancer community and experience represented now in media and cancer non-profits? How do you feel that representation impacts you?	Representation, Peer Connection	Gap
Have you noticed a difference in the care you receive from different medical providers? If yes, what are those differences?	Provider Communication, Mental Health	Gap, Expansion
If you have interacted with Escape, how has Escape impacted your life?	Peer Connection, Creating Space	Asset, Expansion
Is there another Community Based, Cancer and/or LGBTQI2S+ organization that has given you LGBTQI2S+ specific cancer support? In what ways did you receive support?	Peer Connection	Expansion
What would improve your experience with medical providers as a LGBTQI2S+ patient or caregiver?	Provider Communication, LGBTQI2S+ Affirmation	Gap
How can Escape create space for culturally-based approaches to healing? How can Escape advocate for the inclusion and respect of culturally based approaches to healing in medical spaces?	Provider Communication	Gap
Is there anything that we have forgotten to ask that you feel is important for Escape to know about your experience in living with cancer as someone who identifies as LGBTQI2S+?	Emotional Support, LGBTQI2S+ Affirmation	Gap

## Data Availability

Data are available upon reasonable request.

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
