# Peer review of "A Needs Assessment Approach for Adolescent and Young Adult Sexual and Gender Diverse Cancer Survivors"

_ijerph, 2024, doi:10.3390/ijerph21040424_

Round 1

Reviewer 1 Report

Comments and Suggestions for Authors

The present study, “A Needs Assessment Approach for Adolescent and Young

Adult Sexual and Gender Diverse Cancer Survivors,” describes findings from an online needs assessment of Sexual and Gender Diverse (SGD) cancer survivors. This exploratory investigation was completed in partnership with community and academic institutions. Content analysis revealed four major unmet needs related to SGD AYA cancer care (i.e., sexual health and family planning, gender affirmation, financial stability, and emotional support). The needs assessment also highlighted participant perspectives on community, support, and self-care as well as gaps/areas of expansion/assets of a community program (i.e., Escape). Overall, this was a well-written, thorough, and informative manuscript. The authors were comprehensive in their analyses and takeaways. I have very few and minor details that require clarification, as this manuscript was exhaustive. Please see my suggestions below.

-        I would be interested in hearing more about how this expands upon the very limited literature of the discussion. While I find it important to highlight the Why/what/how of Escape, a more comprehensive evaluation of how these findings “fit into” the pre-existing literature or extend upon it would be helpful.

-        While I am not an expert in content analysis, it would be helpful to hear a bit more about the sample size in terms of the analysis. Was the sample large enough to reach thematic saturation/use this technique? It seems like yes, but a sentence stating that would be helpful.

-        I would also add in a sentence or two about the gender/sex breakdown of the sample. Is this breakdown of gender/sex representative of the Sexual and Gender Minority population? If not, it would be helpful to add this into the limitation section as well.

Reviewer 2 Report

Comments and Suggestions for Authors

The research article proposed by Ghazal et al examines the needs of people with cancer in various fields, including gender diversity, and tries to fill this gap in the literature. This is an important aspect because patients with cancer may have different needs during diagnosis and treatment, and their gender diversity should be taken into account. 

I consider examining specific blood factors could better conclude these needs.

The results are well reported, and the references are appropriate.

However, the validity and accuracy of the questionnaires are not mentioned.  Did the author make it? Couldn't you get a correlation with factors related to cancer through questionnaires?

The authors could have a good relationship with their study by studying the reports of people in the hospital and being aware of the laboratory tests, and in this way, the study could have a higher quality.

Reviewer 3 Report

Comments and Suggestions for Authors The manuscript entitled "A needs assessment approach for adolescent and young adult sexual and gender diverse cancer survivors" describes the concept of a community organization dedicated to serving sexual and gender diverse cancer survivors.  

I congratulate the authors on writing this manuscript. However, I have to recommend a few changes to this article.

First, the introduction could be improved to better understand the importance of their research. I suggest a prevalence description

Materials and methods- inclusion and exclusion criteria must be included in this section.

I suggest rewriting some parts of the article due to the high percentage on the ithenticate report.

All in all, this is a well-written article. I suggest accepting after these modifications are made.

Round 2

Reviewer 3 Report

Comments and Suggestions for Authors

The manuscript can be accepted in its current form.

I thank the authors for the changes to the manuscript.